# Structural Design and Finite Element Simulation Analysis of Grade 3 Graded Porous Titanium Implant

**DOI:** 10.3390/ijms231710090

**Published:** 2022-09-03

**Authors:** Bowen Liu, Wei Xu, Mingying Chen, Dongdong Chen, Guyu Sun, Ce Zhang, Yu Pan, Jinchao Lu, Enbo Guo, Xin Lu

**Affiliations:** 1Shunde Graduate School, University of Science and Technology Beijing, Foshan 528399, China; 2National Engineering Research Center for Advanced Rolling and Intelligent Manufacturing, Institute of Engineering Technology, University of Science and Technology Beijing, Beijing 100083, China; 3Institute for Advanced Materials and Technology, University of Science and Technology Beijing, Beijing 100083, China

**Keywords:** titanium, gradient porous structure, oral implant, mechanical properties, biocompatibility

## Abstract

The metal titanium is often used as a dental implant material, and the elastic modulus of solid titanium implants does not match the biological bone tissue, which can easily produce a stress shielding effect and cause implant failure. In this paper, a three-level gradient porous structure implant was designed, and its mechanical and biological adaptability were studied by finite element simulation analysis. Combined with the comprehensive evaluation of the mechanical and biological properties of implants of various structures, the analysis found that a porous implant with porosity of 59.86% of the gradient was the best structure. The maximum equivalent stress of this structure in the mandible that simulated the oral environment was 154.34 MPa, which was less than half of its theoretical compression yield strength. The strain of the surrounding bone tissue lies in the bone compared with other structures, the proportion of the active state of plastic construction is larger, at 10.51%, and the fretting value of this structure and the bone tissue interface is the smallest, at only 10 μm.

## 1. Introduction

With the increasingly serious aging problem, the phenomenon of oral tooth defects has become prominent. Titanium and its alloys have excellent mechanical properties, corrosion resistance, and good biocompatibility, and are often used as oral implant materials [1,2,3,4]. At present, most of the oral titanium implants used on the market are of a fully dense structure. In the initial stage of implantation, fully dense Ti can be better integrated with bone tissue. However, because the elastic modulus of titanium differs greatly from that of biological bone, it is easy to generate a stress shielding effect with bone tissue when occlusal stress is applied [5,6,7]. The stress shielding effect is not conducive to the growth of bone tissue, resulting in implant failure [8]. The design of porous structure reduces the elastic modulus of the implant material, which is close to the elastic modulus of biological bone. Not only that, but the porous structure is conducive to the adhesion and proliferation of bone tissue cells, as well as the transfer of body fluids and nutrients in the pores.

Selective laser melting (SLM) can precisely control the structure and distribution of pores when preparing porous materials, providing the most convenient technical support for the research and development of complex porous titanium [9,10,11,12]. SLM has attracted the broad attention of many materials research scholars. Mullen L. et al. [13] used a simple octahedron as the unit structure with the SLM process to obtain a porous titanium scaffold with a porosity of 10–90%. The compressive yield strength of the porous titanium was between 0.5–350 MPa. After a heat treatment process, the compressive yield strength of SLM-Ti was increased to 40–56.4 MPa, the elastic modulus was 3.5–6.5 GPa, and the porosity reached 65%. Peng et al. [14] used the same method to design and prepare hexagonal single porous titanium alloy parts with a pore size of 700 μm and porosity of 81.23%. The sample’s elastic modulus was 2.23 GPa, and its compressive strength was 22.57 MPa. According to the research results of these scholars, it was found that the elastic modulus of porous titanium materials was greatly reduced. However, when the porosity of a porous titanium alloy with single pore structure is large, its strength is small. The mechanical properties of a single porous structure can not meet the mechanical strength requirements during implanting. This leads to deformation and fracture easily under load in the actual implant environment. In this paper, a gradient porous structure titanium implant was designed. The principle of gradient porous structure design is that the outer layer of the porous implant has the characteristics of high porosity and large pore size. This structure can meet the space environment conditions required for osteoblast adhesion and generation, improve the bone integration ability, and make the implant structure have good bone conduction ability [15,16,17,18]. The inner structure of the implant has low porosity so that the strength of the overall structure is improved to meet the mechanical performance requirements of the implant.

Therefore, in the design of the gradient structure of the implant, it is necessary to consider its mechanical properties and its biological adaptability. However, the current evaluation of gradient porous implants is mostly focused on their mechanical properties. Only a few researchers have performed research on the biological adaptation of gradient porous implants in a specific biological environment. Therefore, the increase in gradient porous structure implant research on biological adaptability can be expected to provide a certain measure of support and reference value to the research field of dental implants. In order to facilitate and quickly study the biological fit between the gradient porous structure and bone tissue, the finite element method is widely welcomed by researchers in analyzing the stress distribution of oral implants and surrounding bones. Additionally, the finite element method is considered to be a reliable and accurate tool for analyzing the mechanical behavior of prostheses [19,20]. Instrumental studies such as the finite element method are an excellent tool to evaluate anatomical structures and any rehabilitation facilities before conducting animal experiments in order to have mechanical properties and satisfactory load cycle tests. Using ANSYS Workbench numerical simulation software, the implants with different pore structures are placed in a simulated oral environment. The stress and strain distribution of the implant and surrounding bone tissue and the micro-movement of the implant-bone tissue interface are evaluated, and the influence of the gradient structure on the mechanical properties and biological adaptability of the implant is obtained to meet the requirements of the implant with the best gradient porous structure.

## 2. Results and Discussion

### 2.1. Mechanical Properties of Columns with Different Gradient Porous Structures

Figure 1 shows the corresponding relationship between the column pillars with gradient porous structure and porosity. When the width of the pillars is 0.1 to 0.3 mm, the porosity of the pillars with a porous structure is 45% to 92.64%. It can also be found from the figure that the porosity has a linear functional relationship with the pillar width, and the relationship is shown in Formula (1).
P = −238.8w + 118.6(1)

In the formula, P is the porosity of the gradient porous structure column, in %. w is the width of the gradient porous structure column, in mm.

In the process of designing the structure in the future, the porosity of the gradient porous structure column can be controlled by adjusting the pillar width according to this formula to obtain the desired structure.

It can be seen from Figure 2 that the equivalent stress distribution of each gradient porous structure column is relatively uniform, and there is no large stress concentration area. The maximum equivalent stress appears in the middle area of the upper surface of the pillar. As the porosity increases, the maximum equivalent stress on the gradient porous structure column increases. The lower the porosity of the porous structure, the better the compression performance. It can be seen from Figure 3a that, except for the structure with a porosity of 92.64%, its maximum equivalent stress exceeds the compressive yield strength of titanium material by 607 MPa. The maximum equivalent stress of the other four structures is lower than the compression yield strength to ensure the normal use of the structure. Additionally, when the porosity of the gradient porous structure column is 45% and 59.86%, the maximum equivalent stress is much lower than the compressive yield strength of the titanium material.

According to the Gibson–Ashby model and related theories [21], the corresponding elastic modulus E and the theoretical yield strength σ of the gradient structure implant are calculated:E = E_s_ (1 − φ)^2^(2)
σ = σ_s_ (1 − φ)^3/2^(3)

Among them, E_s_ is solid titanium elastic modulus (110 GPa), σ_s_ is solid titanium compressive yield strength (607 MPa), and φ is porosity.

Using Formula (2) to calculate the elastic modulus value of the gradient porous structure column with different porosity, the result is shown in Figure 3b. From the figure, when the porosity is 59.86–83.99% of the structure, the elastic modulus value is close to that of biological bone.

### 2.2. The Influence of Porosity on the Mechanical Properties of Implants

Static simulation analysis was performed on the implant-mandibular model. The equivalent cloud diagram and the maximum equivalent stress line diagram of the implants with different porosities under the action of the vertical stress in the simulated oral environment are shown in Figure 4. It can be seen from Figure 4f that as the porosity increases, the maximum equivalent stress on the implant gradually increases. It can be seen from the equivalent stress cloud diagram of the implant in Figure 4a–e that the maximum equivalent stress of each structure appears in the contact area between the solid part of the implant and the porous part, which is in line with the actual law. Because the structure of this area changes greatly, stress concentration is likely to occur. When the porosity of the pillar part of the gradient porous structure of the implant is 45% and 59.86%, the maximum equivalent stress of the implant is less than 50% of its theoretical yield strength. At this time, the mechanical performance requirements of oral implants can be met.

In order to facilitate the verification of the accuracy of the compression yield strength and elastic modulus of the gradient porous structure implant cylinder, the Abaqus simulation software was used to perform static compression simulations on different gradient porous structures, and the compression stress–strain simulation curves of each structure were obtained, as shown in Figure 5a. By analyzing the compressive stress–strain simulation curve, the elastic modulus of each structure and its compressive yield strength can be obtained.

It can be seen from Figure 5b that as the porosity increases, the slope of the curve becomes smaller, that is, the elastic modulus value of the structure decreases. Comparing with Figure 3b, it is found that the elastic modulus value of each structure calculated by Abaqus simulation software is slightly larger than the elastic modulus value obtained by using the Gibson–Ashby theoretical formula. This is because the Gibson–Ashby theoretical model is based on the porosity distribution. Uniformity is the premise, and the porosity of our model is distributed in a gradient. The internal porosity is low, and the elastic modulus value of the overall structure is slightly increased in the calculation. Although the elastic modulus value of each structure obtained by simulation is slightly larger, it is still close to the elastic modulus of biological bone.

It can be seen from Figure 6 that the compressive yield strength values of each structure calculated using the Abaqus simulation software and the Gibson–Ashby theoretical model formula are very small. When the porosity is 59.86%, the yield strength values calculated with the two methods are very close. Additionally, it can be seen from the figure that when the porosity of the gradient porous structure is 59.86%, the compressive yield strength of the material exceeds 150 MPa. Although the porosity is close, the compressive yield strength is greater than that in the literature [13]. The mechanical behavior of the designed gradient porous structure column can be effectively predicted, which is convenient for subsequent testing and screening of the mechanical properties of actual specimens.

### 2.3. The Effect of Porosity on the Biological Fit of Implants

In this study, biocompatibility was evaluated based on the the equivalent stress of the implant and its surrounding bone tissue, and the micro-movement of the bone tissue interface between the implant and the contact part. The implant is subjected to chewing force in the oral environment and transfers the force to the surrounding bone tissue. If the bone tissue receives too little stress for a long time, it will lead to bone resorption. At the same time, too much stress should be avoided to prevent the occurrence of pathological fractures. Therefore, the stress transmitted by the implant to the surrounding bone tissue cannot be too small or too large. According to Frost’s minimum effective strain theory [22], the effect of stress and strain on bone tissue has four thresholds. If the stress of the bone tissue is less than 1–2 MPa (strain is less than 50–100 με), the bone resorption rate is greater than the reconstruction speed, and the bone tissue is absorbed as waste; when the stress is in the range of 2–20 MPa (strain is 100–1500 με), the bone formation speed and absorption speed are roughly the same, maintaining normal bone quality and increasing appropriately; when the stress is in the range of 20–60 MPa (strain is in the range of 1500–3000 με), it is in the bone plastic. In an active state, bone stress within this range can promote bone tissue growth; when the stress is in the range of 60–120 MPa (strain is in the range of 3000–25,000 με), micro-damages are accumulated in the bone tissue. In addition, the main contact part of the implant implanted in the bone tissue is cancellous bone, so under load, the proportion of cancellous bone within 1.5 mm around the implant in the active state of bone reconstruction (in the range of 1500–3000 με) is calculated to evaluate the biological fit of the implant. It can be seen from Figure 7 that as the porosity of the implant increases, the maximum equivalent strain of the surrounding cortical bone increases. However, when the maximum equivalent strain value of the cortical bone around the implant with a porosity of 92.64% is in the pathological overload state of the bone tissue (the strain is in the range of 3000–25,000 με, the bone tissue accumulates micro-damage), and other structures are implanted, the microvariation values of cortical bone around the body are all within the range of proper growth state and active state of bone plastic parts.

Figure 8a–e is the equivalent strain cloud diagram of the cancellous bone around the implant with different porosity, and Figure 9f is the equivalent strain line graph of the cancellous bone around the implant with different porosity. As the porosity of the body increases, the maximum equivalent strain of the cancellous bone around each implant decreases first and then increases. When the porosity of the implant is 59.86% and 72.87%, the maximum equivalent strain of the surrounding cancellous bone value is relatively low. This is related to the structure of the implant and the overall mechanical characteristics. Because the elastic modulus of the structure with porosity of 59.86% and 72.87% is close to that of the surrounding bone tissue, the size of the outermost pore is relatively moderate, which is effective in the case when the stress on the surrounding bone tissue is closer to the outside of the implant. The maximum equivalent strain on the bone tissue is relatively small at this time. It can be seen from Figure 8g that the average equivalent strain of the cancellous bone around the implant of each structure is consistent with the maximum equivalent strain. The average equivalent strain of the cancellous bone around the implant of each structure is in the bone within the strain value range of proper growth state. Figure 8h shows the proportion of the strain value of the cancellous bone around the implant with different porosity in the range of 1500–3000. It can be found that the porosity of the gradient porous structure column of the implant is 59.86% and 92.64%. At this time, the equivalent strain of cancellous bone around the implant has the highest proportion in the range of 1500–3000, which is more beneficial to the growth of surrounding bone tissue. Figure 8i shows the dispersion of the equivalent strain of the cancellous bone around the implant with different porosity. It can be found that when the porosity is 59.86%, the dispersion is the smallest at this time, indicating that the data distribution is more uniform and stable.

Some scholars believe that the fretting value of the implant–bone tissue interface also has a significant impact on the stability of the implant. Brunski et al. [23] showed that in order to achieve osseointegration rather than fibrous healing, the fretting value of the implant-bone tissue should be less than 100 µm. Trisi et al. [24] believed that the fretting value of implant-bone tissue should not exceed 50–100 µm, so as to ensure the long-term stability of the implant. Figure 9 is a broken line diagram of the interface micro-motion values between implants with different pore structures and the surrounding cancellous bone. It can be seen from the figure that with the increase in porosity, the fretting value of the interface between implant and cancellous bone first decreases and then increases. It is generally believed that the micro-motion value of the implant-bone tissue should not exceed 50–100 µm to promote the formation of bone tissue surrounding the implant instead of bone fiber surrounding the implant. When the porosity of the implant gradient porous structure column is 59.86%, the interface fretting value is the smallest, only about 10 µm, which is much less than 50 µm. Therefore, the implant with this structure is more stable at the initial stage of implantation in the bone tissue. It facilitates the reliability of implant placement, and it can be found in the figure that when the porosity is 92.64%, the implant–bone tissue interface fretting value is the largest, at about 50 µm, and the fretting values of implant–bone tissue interface of other structures are all less than 50 µm, possessing osseointegration ability after implant placement.

To sum up, the optimal structural parameters of the implant gradient porous structure pillar are porosity of 59.86% and pillar width of 0.25 mm. The maximum equivalent stress of the structure is much lower than the compression yield of titanium material strength, at this time. Its elastic modulus is closer to that of biological bone. The compressive yield strength value calculated by the Gibson–Ashby theoretical model and Abaqus exceeds 150 MPa, which is higher than that reported in the literature [13] (56.4 MPa). It is beneficial to meet the needs of its mechanical strength. When simulating and analyzing the equivalent strain of the bone tissue, the equivalent strain value of the bone tissue around this structure accounts for a higher proportion of the bone structure in the active state, and the fretting value of the implant and the surrounding bone tissue of this structure is the smallest, which is conducive to the success of implantation.

## 3. Materials and Methods

### 3.1. Built Model

In this paper, we first design a suitable fan structure basic unit and build a gradient porous structure implant on this basis. Considering the suitable pore size range for bone ingrowth, combining the research of various scholars [7,24,25], when the pore size range is 400–800 μm, it is beneficial to the ingrowth of osteoblasts. Therefore, the basic unit of the sector structure with the outermost pore size in this range is designed. The structural model of the basic unit of each sector structure and the gradient porous structure column are shown in Figure 10a. The structural parameters of the basic unit of each sector structure are shown in Table 1.

The modeling software of this paper is Solidworks, Dassault Systemes Co., Ltd., Concord, MA, USA. Using Solidworks modeling software, the gradient porous implant model and the mandible model of the implant environment were constructed. The implant part is a gradient porous structure, with a length of 10 mm and a diameter of 4 mm. The neck height is 1.8 mm, the upper diameter is 4.8 mm, and the lower bottom diameter is 4 mm. A 5 mm high abutment is designed on the upper part of the implant. The top diameter of the abutment is 3 mm and the bottom diameter is 4 mm, which is simplified as a whole with the implant. The bone tissue mainly in contact with the gradient structure of the implant is cancellous bone, and the bone tissue within 4.2 mm from the implant surface is the main stress-affected area [24]. So, the mandibular bone block model is simplified to a total height of 15.5 mm, mesiodistal length of 12.4 mm, buccal-lingual length of 12.4 mm, and outer cortical bone thickness of 1.3 mm. The model is shown in Figure 10b,c.

### 3.2. Statics Simulation

The finite element analysis software used in this experiment is ANSYS Workbench 18.0, ANSYS Co., Pittsburgh, PA, USA. The basic parameters of the Ti material used are based on the characteristics of the research team’s previous research: density 4.46 g·cm^−3^, elastic modulus 110 GPa, compressive strength 819 MPa, tensile strength 894 MPa.

#### 3.2.1. Material Parameter Setting

The use of three-dimensional finite element analysis for biomechanical analysis requires the simplified processing of complex human tissues and materials. The implant material is titanium. The mandibular model includes cancellous bone and cortical bone. The relevant parameters of bone tissue and implant materials are shown in Table 2 [26].

#### 3.2.2. Meshing

The implants in the oral environment are divided into tetrahedral meshes. The 1.5 mm thickness of the implant–bone tissue interface uses a denser mesh. The mesh size of this part is set to 0.3 mm, and the mesh size of other parts is set to 0.5 mm, and the models of each group are consistent.

#### 3.2.3. Contact and Constraint Setting

The implant body and neck, implant neck and abutment, and cortical bone and cancellous bone are all set to be in binding contact. When analyzing the maximum equivalent stress and maximum equivalent strain of the bone tissue, it is assumed that complete osseointegration occurs between the implant and the surrounding bone. Therefore, the binding contact between the implant and the surrounding bone is set as a binding contact without sliding friction. When calculating the micro-movements of the implant–bone tissue interface, it is assumed that after the implant is stressed, there will be compression between the implant and the bone interface, and there will be a slight sliding along with the interface, and the friction coefficient is set to 0.3. In the simulation process, the buccal-lingual surface, mesiodistal surface, and bottom surface of the bone block model are set as rigid constraints, that is, it is assumed that the mandible does not move and does not shift.

#### 3.2.4. Loading Method

This study has shown that after the implant is implanted in the jaw, the average maximum bite force of the first premolars and molars is 200 N [27,28]. Therefore, a 200 N vertical load was used to load the implant-mandible model.

#### 3.2.5. Evaluation of Calculation Structure

This study used ANSYS Workbench software to perform static analysis on the model, obtain the stress and strain distribution cloud diagram of the implant and the bone tissue around the implant, calculate the stress dispersion of the implant and the cancellous bone, and analyze the distribution of the equivalent strain interval of the cancellous bone. Stress dispersion is defined as the ratio of the width of the stress distribution to the average stress. The smaller the dispersion, the more uniform the stress distribution. The calculation of fretting is to define a node on the surface of the implant and determine the node at the corresponding position of the bone interface. After the loading force is applied, the relative displacement between the two nodes on the *x*-axis, *y*-axis, and *z*-axis of the three-dimensional coordinate system is calculated. Therefore, when measuring implant micromotion, reference points are taken on the neck, body, and end of the implant to measure the buccal-lingual (*x*-axis), vertical (*y*-axis), and near-distal (*z*-axis) directions. The displacements of the axis are dx_1_, dy_1_, dz_1_, and the displacements of the corresponding points on the bone tissue interface are measured at the same time as dx_2_, dy_2_, dz_2_. According to Formula (4), the comprehensive relative displacement, that is, the fretting value, is calculated as:S = √((dx_1_ − dx_2_)^2^ + (dy_1_-dy_2_)^2^ + (dz_1_ − dz_2_)^2^)(4)

## 4. Conclusions

(1) With the increase in porosity, the maximum equivalent strain of the implant gradient porous structure column gradually increases. The lower the porosity, the better the compressive performance. When the porosity is 45–83.99%, the maximum equivalent stress value of the gradient porous structure column is less than the compressive yield strength of the titanium material. In the simulation of oral environment statics, the maximum equivalent stress on the overall structure of the implant increases with the increase in porosity. When the porosity is 45% and 59.86%, the maximum equivalent stress on the implant is now less than 50% of the theoretical yield strength of its structure, meeting the requirements of stomatology for the mechanical properties of implants.

(2) With the increase in porosity, the equivalent strain value of cortical bone increases, and the maximum equivalent strain value and average equivalent strain value of cancellous bone first decrease and then increase. When the porosity is 59.86%, the strain data of cancellous bone around the implant accounts for the highest proportion in the range of 1500–3000, which is more conducive to active bone reconstruction. However, when the porosity is 92.64%, the maximum equivalent stress is too large, and it is easy to cause pathological fractures in the bone tissue, which is not conducive to implants.

(3) The optimal structural parameters of the implant gradient porous structure pillar are porosity of 59.86% and pillar width of 0.25 mm. The structure receives the maximum equivalent stress from the implant gradient porous structure pillar and overall, under the comprehensive evaluation of the maximum equivalent stress of the implant and the strain of the surrounding bone tissue and the micro-movement of the implant–bone tissue interface. Its performance is the best, which can meet the needs of the implant and is beneficial to the growth of the surrounding bone tissue. Reproduction is more beneficial to the long-term stability of the implant after implanted in the bone tissue. It is worth pointing out that this paper is mainly based on the optimal structure calculated by finite element simulation. Specific biological experiments need to further explore the matching between experiments and simulation.

## Figures and Tables

**Figure 1 ijms-23-10090-f001:**
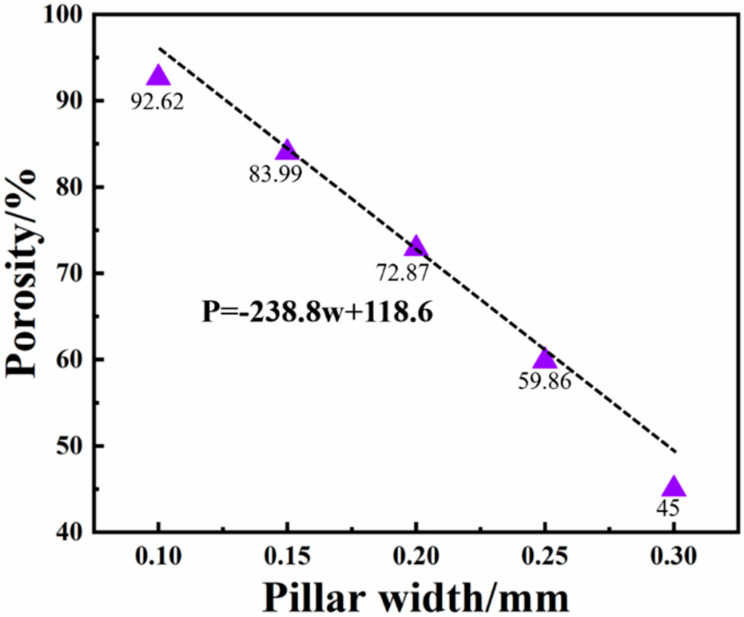
Correspondence between column width and porosity of gradient porous structure.

**Figure 2 ijms-23-10090-f002:**
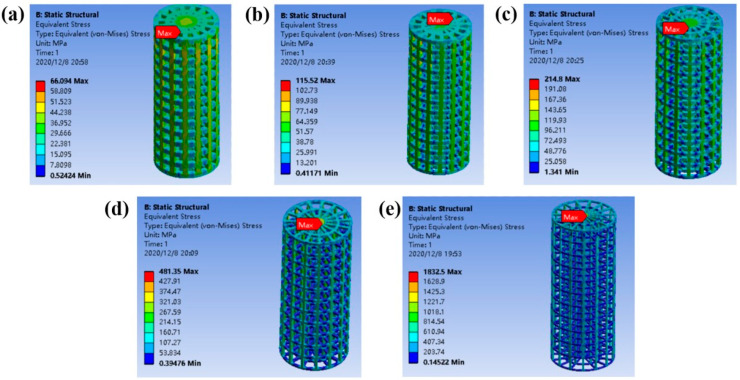
Static equivalent stress cloud diagram of gradient porous structure cylinders with different porosities: (**a**) Porosity 45%, (**b**) Porosity 59.86%, (**c**) Porosity 72.87%, (**d**) Porosity 83.00%, (**e**) Porosity 92.64%.

**Figure 3 ijms-23-10090-f003:**
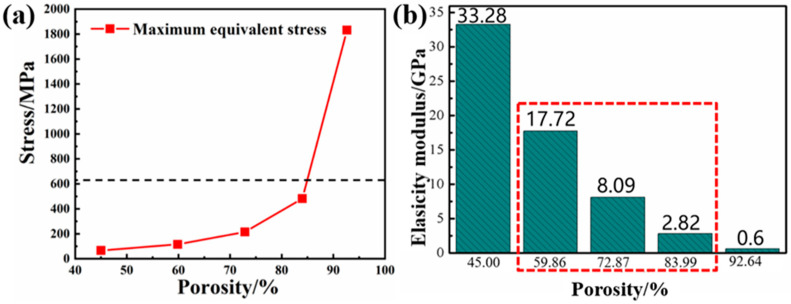
(**a**) Line diagram of the maximum equivalent stress experienced by a gradient porous structure column with different porosity. (**b**) Elastic modulus of gradient structure column with different porosity.

**Figure 4 ijms-23-10090-f004:**
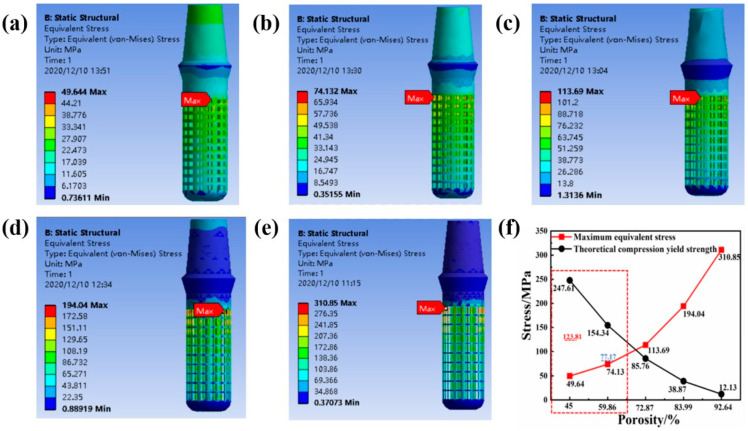
Equivalent stress cloud diagram and maximum equivalent stress line diagram of implants with different porosity under the action of 200 N vertical stress. (**a**) Porosity 45%, (**b**) Porosity 59.86%, (**c**) Porosity 72.87%, (**d**) Porosity 83.00%, (**e**) Porosity 92.64%. (**f**) The broken line diagram of the maximum equivalent stress and theoretical yield strength of each structure.

**Figure 5 ijms-23-10090-f005:**
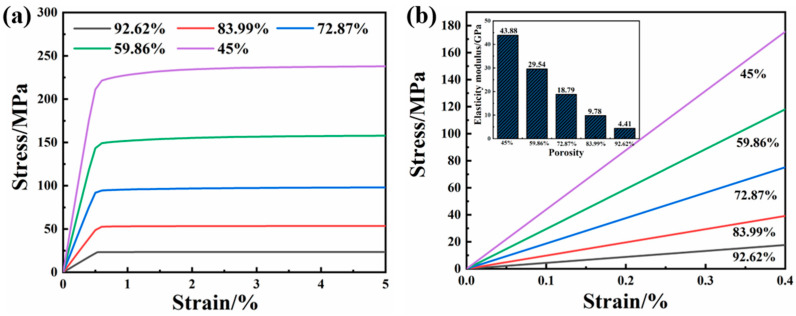
(**a**) Compressive stress–strain simulation curves of different structural columns. (**b**) The simulated elastic modulus of porous structures with different porosity gradients.

**Figure 6 ijms-23-10090-f006:**
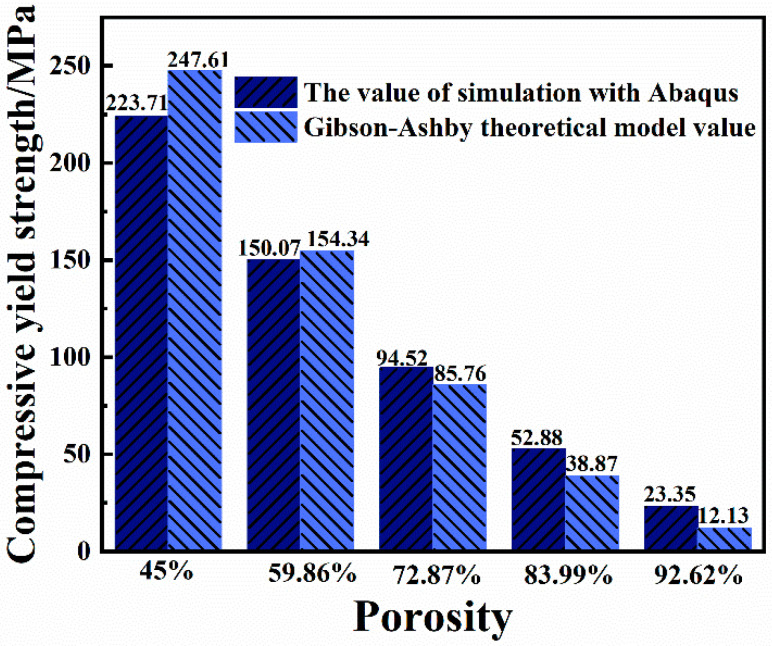
The compressive yield strength of each structure obtained by simulation calculation and formula calculation.

**Figure 7 ijms-23-10090-f007:**
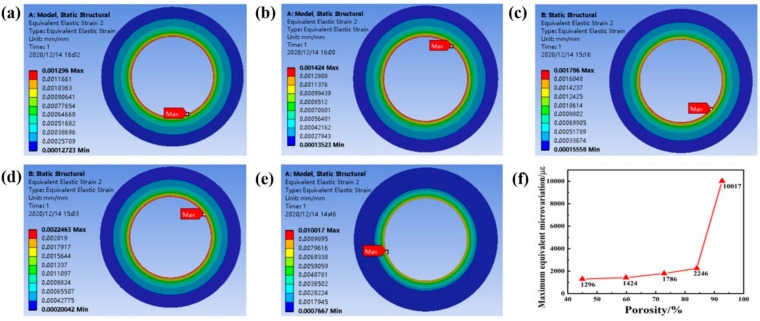
(**a**–**e**) Porosity 45–92.62%, equivalent strain cloud diagram and maximum equivalent strain line diagram of cortical bone around implants with different porosity under 200 N vertical stress. (**f**) The line graph of the maximum equivalent strain on the cortical bone near the implant of each structure.

**Figure 8 ijms-23-10090-f008:**
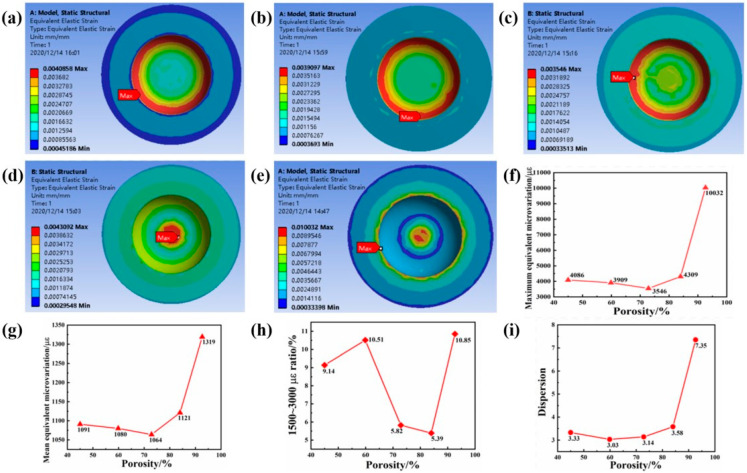
(**a**–**e**) Porosity 45–92.62%, equivalent strain cloud diagram of cancellous bone under 200 N compressive stress, (**f**) maximum equivalent strain, (**g**) average equivalent strain, (**h**) proportion of 1500–3000, (**i**) effect variation dispersion.

**Figure 9 ijms-23-10090-f009:**
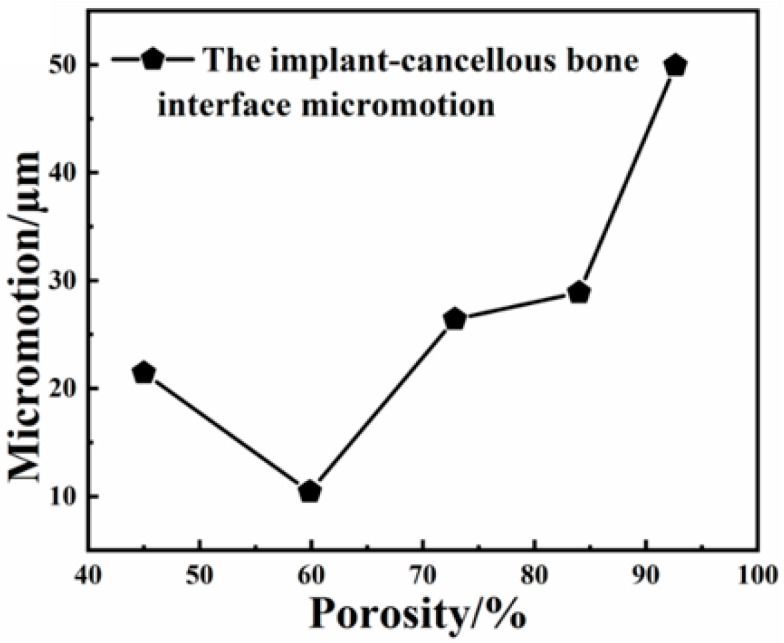
The micro-motion values of the implant-cancellous bone interfaces of each structure.

**Figure 10 ijms-23-10090-f010:**
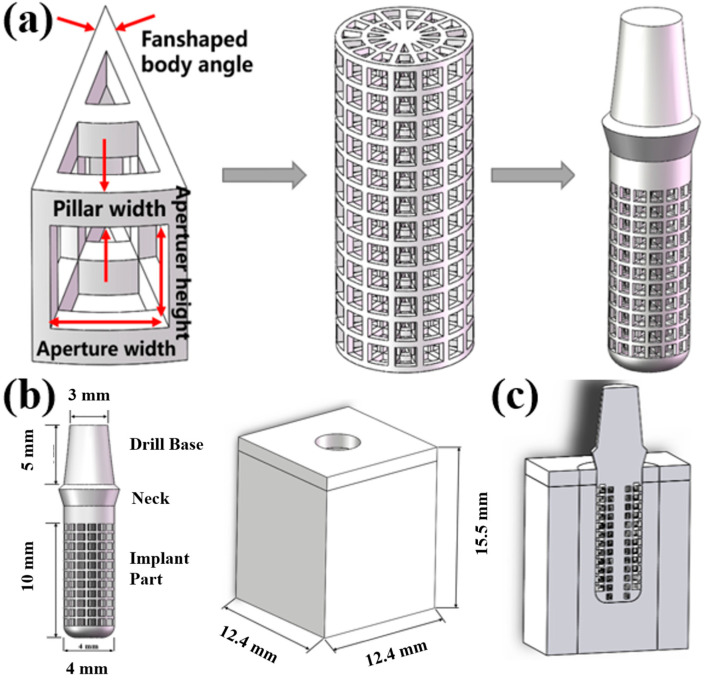
Three-dimensional model of implant and mandible: (**a**) Implant model creation process. (**b**) Implant and mandible model size parameters. (**c**) Sectional view of implant-mandible mode.

**Table 1 ijms-23-10090-t001:** Basic unit model parameters of each sector structure.

Pillar Width/mm	Fan-Shaped Body Angle/°	Aperture Height/mm	Aperture Width/mm
0.1	24	0.8	0.73
0.15	24	0.7	0.68
0.2	24	0.6	0.63
0.25	24	0.5	0.59
0.3	24	0.4	0.54

**Table 2 ijms-23-10090-t002:** Related parameters of implant-bone tissue material settings.

Material	Elastic Modulus/GPa	Poisson’s Ratio
Titanium	110	0.35
Cortical bone	13.7	0.3
Cancellous bone	1.37	0.3

## Data Availability

Not applicable.

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
