# Peer review of "Structural Design and Finite Element Simulation Analysis of Grade 3 Graded Porous Titanium Implant"

_ijms, 2022, doi:10.3390/ijms231710090_

Round 1

Reviewer 1 Report

Dear editor

This manuscript is well designed. The idea of this MS is interesting for implant dentistry. M and M looks solid and analysis setting is well managed. However, manuscript structure is not easy to read. M and M should be placed aftre Introduction. I do not have any negative comments. 

Author Response

We greatly appreciate the very constructive comments received and would like to sincerely thank the reviewers and editor for their significant investment of time in reviewing our manuscript.

Special thanks for your very constructive comments. This positive comment gives us huge encouragement for this work. 

Reviewer 2 Report

I have reviewed the manuscript, I found this work impactful and fit well with in the scope of this journal. The manuscript needs some major improvements; there are a few suggestions that authors may consider to improve it further: 

The use of the English language is reasonable, however, there are a number of punctuation and grammatical errors; that should be corrected and rephrased using academic English for a better flow of text for reader.

Abstract: is precisely written, and the aim of the study is mentioned. Please include some more information about the results/finding to enhance the impact of this section.

The introduction; is detailed, compact, covering the background information and the rationale of the study effectively. However, the last paragraph is very details and suggested to condense that information, not all readers have the knowledge of FEM method, I would like to suggest this paper to be included to improve your manuscript: 

"Application of bioengineering devices for stress evaluation in dentistry: The last 10 years FEM parametric analysis of outcomes and current trends" DOI

10.23736/S0026-4970.19.04263-8.

Materials and methods- this section is well organized, 

Please, describe this section  meticulously as this is very important for the readers. 

Furthermore, In discussion, more studies in context should be included; as there is little support of literature from the previous studies.

In the section conclusion please write the limits of present study, because it's an in vitro experiment. 

I believe that your manuscript would have much more relevance after suggested improvements.

Author Response

We greatly appreciate the very constructive comments received and would like to sincerely thank the reviewers and editor for their significant investment of time in reviewing our manuscript. In addition, we have rectified a number of  grammatical and syntax errors. We would be happy to further revise this manuscript if needed. Thank you.

According to your suggestion, some introduction about finite element simulation is added. Due to space constraints, it is not possible to expand on the details.

In the conclusion part, this paper adds the discussion of the remaining limitations of this paper.

Reviewer 3 Report

The manuscript makes for heavy reading due to grammatical errors and long sentences. Please try to make shorter sentences to make them easy for the reader. There are many hanging sentences and the use of capital letters inappropriately.

I miss any mention of validity and accuracy testing of the model. Please include how you tested your model to be accurate. 

Author Response

We greatly appreciate the very constructive comments received and would like to sincerely thank the reviewers and editor for their significant investment of time in reviewing our manuscript. In addition, we have rectified a number of grammatical and syntax errors. We would be happy to further revise this manuscript if needed. Thank you.

Reviewer 4 Report

Dear authors,

The paper is associated with the interesting topic of developing new materials and designs for dental implants in order to improve the biocompatibility, the mechanical properties and the biologic response of the host tissue. In my opinion, the article can be published after solving the following issues:  

1. The manuscript is very difficult to read and follow because of many writing and grammar errors, so I strongly recommend to use the extensive editing of English language. 

Here are some examples:

- Lines 29-30 please change oral tooth defect with a more appropriate medical term

- Lines 33-34 please rephrase and avoid to repeat bone tissue, also in the following lines  

- Line 37 there is a coma and then an uppercase

- Line 38 please use the article: a porous implant

- Line 50 uppercase after coma

- Line 58 please change the word small with a more appropriate one

- Line 82 please use the plural

2. Please edit the References changing the abbreviations with the appropriate authors´ names

3. Please specify the details of the software: the producer, the country...

Author Response

(The authors gave the same response as above.)
